# A Multiplex PCR Assay Combined with Capillary Electrophoresis for the Simultaneous Identification of Atlantic Cod, Pacific Cod, Blue Whiting, Haddock, and Alaska Pollock

**DOI:** 10.3390/foods10112631

**Published:** 2021-10-29

**Authors:** Yu-Min Lee, Shinyoung Lee, Hae-Yeong Kim

**Affiliations:** Institute of Life Sciences & Resources, Department of Food Science & Biotechnology, Kyung Hee University, Yongin 17104, Korea; lym5373@naver.com (Y.-M.L.); shinyoung16@gmail.com (S.L.)

**Keywords:** cod, multiplex PCR, species identification, capillary electrophoresis

## Abstract

With an increased consumption of seafood products, food fraud with fish resources has been continuously reported. In particular, codfish has been exploited worldwide as a processed product in fresh, frozen, smoked, canned, or ready-to-eat dish forms. However, it is challenging to identify processed fish products after processing because of their similar morphological characteristics. Substitution and mislabeling of codfish among different species are also happening deliberately or unintentionally. Thus, it is necessary to distinguish cod species to prevent fish adulteration and food fraud. In this study, we developed a multiplex PCR for simultaneously identifying five cod species within *Gadidae* using capillary electrophoresis. Then, their species-specific primer sets were designed by targeting the mitochondrial *cytochrome b* gene. Subsequently, the amplicon sizes obtained were 237 bp, 204 bp, 164 bp, 138 bp, and 98 bp for Atlantic cod, Pacific cod, blue whiting, haddock, and Alaska pollock, respectively. The specificity of each primer was further tested using 19 fish species, and no cross-reactivity was observed. The limit of detection of this multiplex PCR assay was 1 pg. The developed multiplex PCR assay can be applied to 40 commercial food products successfully. This detection method will be efficient for managing seafood authentication by simultaneously analyzing multiple cod species.

## 1. Introduction

As consumers’ interests in health and nutrition are increasing, the consumption of seafood is on the rise due to its abundance of high-quality protein, n-3 unsaturated fatty acids, and minerals [1]. The growing global fish market and processed fish products lead to deliberate or intentional species substitution of fish and their products. Seafood fraud, such as species substitution, adulteration, and mislabeling, has become a crucial problem globally [2]. Thus, species identification is critical for accurate labeling and resolution of economic issues [3].

The *Gadidae* family is a valuable and commercially important fish resource because cod, which is a member of this family, is the most consumed marine fish containing abundant vitamins and omega-3 fatty acids (i.e., EPA and DHA) [4]. Atlantic cod (*Gadus morhua*), Pacific cod (*Gadus macrocephalus*), blue whiting (*Micromesistius poutassou*), haddock (*Melanogrammus aeglefinus*), and Alaska pollock (*Gadus chalcogrammus*) belong to the *Gadidae* family. However, as codfish is mainly processed in different forms, such as frozen, smoked, and canned, and indicates similar morphological and organoleptic properties after processing, it is difficult to distinguish those species by examining their visual features alone [4]. Furthermore, the price of each species is different, which indicates that the substitution of specific species with cheaper ones can cause economic problems. For instance, it is reported that the price of Atlantic cod is comparatively higher than that of the other *Gadidae* species (i.e., haddock and Alaska pollock) [5]. Likewise, several studies have confirmed cases of Atlantic cod being substituted with Pacific cod, Alaska pollock, or haddock [6,7,8,9]. Additionally, the error rate of cod labeling ranged from 0% to 60% in nine North Atlantic countries, and 7.4% and 28.4% labeling errors were found from 226 cod products in the UK and Ireland, respectively [10]. Hence, developing an accurate method for identifying *Gadidae* species to prevent adulteration and mislabeling of codfish products is required.

Compared with morphology-based identification methods, DNA-based analysis is useful for identification of fish species [11,12,13]. This method is preferable because DNA is more stable under high temperatures and pressure conditions compared with protein. Thus, DNA-based detection methods can be applied to processed food products. In particular, PCR is one of the most popular molecular detection methods used for species identification. Among various PCR-based methods, such as conventional PCR, real-time PCR, ultrafast PCR, and loop-mediated isothermal amplification (LAMP), multiplex PCR is simple and fast for simultaneous identification of multiple species at low cost [14,15,16,17]. Moreover, as capillary electrophoresis offers better resolution results than agarose-based electrophoresis when DNA amplicons are distinguished, we developed a rapid and sensitive multiplex PCR method coupled with capillary electrophoresis to detect Atlantic cod, Pacific cod, blue whiting, haddock, and Alaska pollock species, simultaneously. Afterward, this developed method was applied to commercial food products to verify its applicability under various processed conditions.

## 2. Materials and Methods

### 2.1. Samples

Five *Gadidae* species, Atlantic cod (*Gadus morhua*), Pacific cod (*Gadus macrocephalus*), blue whiting (*Micromesistius poutassou*), haddock (*Melanogrammus aeglefinus*), and Alaska pollock (*Gadus chalcogrammus*) were obtained from the National Institute of Food and Drug Safety Evaluation as a single reference, and each of them was used as the reference. Moreover, 14 non-*Gadidae* species, including common carp (*Cyprinus carpio*), leather carp (*Cyprinus carpio nudus*), goldfish (*Carassius auratus*), Chinese muddy loach (*Misgurnus mizolepis*), snakehead (*Channa argus*), Nile tilapia (*Oreochromis niloticus*), Pacific saury (*Cololabis saira*), Pacific chub mackerel (*Scomber japonicus*), longtooth grouper (*Epinephelus bruneus*), convict grouper (*Epinephelus septemfasciatus*), Atlantic salmon (*Salmo salar*), masu salmon (*Oncorhynchus masou*), swordfish (*Xiphias gladius*), and Patagonian toothfish (*Dissostichus eleginoides*) were obtained from the three local markets in Korea, and the purchased samples from different markets were mixed before use. A total of 40 commercial products were purchased from the markets directly or online as a single item. Different types of processed products, such as dried, egg, minced, cake, roasted, salted, cutlet, boiled, fillet, and fried foods were included. Afterward, all the samples were washed with distilled water, cut into small pieces, and stored at −20 °C until use.

### 2.2. DNA Extraction

DNA was extracted from these 19 fish samples and 40 processed products using a DNeasy Blood & Tissue Kit (Qiagen, Hilden, Germany) according to the manufacturer’s instructions. The concentration and purity of DNA samples estimated using a Maestro Nano-spectrophotometer (Maestrogen, Las Vegas, NV, USA). Only the pure DNA samples, which indicated purity levels between 1.8 and 2.0, were used for further studies.

### 2.3. Primer Design

Sequences of the mitochondrial cytochrome b (*cytb*) genes for the 19 fish species were obtained from the National Center for Biotechnology Information (NCBI, http://www.ncbi.nlm.nih.gov accessed on: 25 June 2021.). The reference sequences were aligned using the Clustal Omega alignment system (http://www.ebi.ac.uk/Tools/msa/clustalo/ accessed on: 25 June 2021), then specific regions were targeted to design primers of each species. Primers were designed by the Primer Designer version 3.0 (Scientific and Educational Software, Durham, NC, USA) software, and synthesized using Bionics (Seoul, Korea). Sequences of the species-specific primer sets used are shown in Table 1.

### 2.4. Specificity and Sensitivity of Simplex PCR

The specificity of each primer set was evaluated using DNA extracted from the 19 fish species, including Atlantic cod, Pacific cod, blue whiting, haddock, and Alaska pollock. Their sensitivities were also estimated with 10-fold serially diluted DNA ranging from 10 ng to 0.01 pg.

Simplex PCR reaction was conducted using a 25 μL reaction volume mix containing 2.5 μL of the 10× buffer (Bioneer, Daejeon, Korea), 800 μM dNTPs (Bioneer), 0.5 unit of the Hot Start *Taq* DNA polymerase (Bioneer), 400 nM of each primer, and 10-ng template DNA. Afterward, the reaction was performed in a thermal cycler (Astec, Tokyo, Japan) under the following conditions: predenaturation at 95 °C for 5 min, 40 cycles of denaturation at 95 °C for 30 s, annealing at 58 °C for 30 s, and extension at 72 °C for 30s, all followed by the final extension at 72 °C for 5 min. Finally, capillary electrophoresis was used to confirm PCR products using an Agilent Bioanalyzer (Agilent Technologies, Santa Clara, CA, USA) and a DNA 1000 Lab Chip kit (Agilent Technologies).

### 2.5. Specificity and Sensitivity of Multiplex PCR

The specificity of the multiplex PCR reaction was tested with 10 ng of genomic DNA samples isolated from the 19 fish species. The sensitivities of these samples were then measured using genomic DNA from 10 ng to 0.01 pg. Subsequently, the multiplex PCR reaction process for the simultaneous identification of Atlantic cod, Pacific cod, blue whiting, haddock, and Alaska pollock was conducted with a 25 μL reaction volume mix, containing 2.5 μL 10× buffer (Bioneer, Daejeon, Korea), 800 μM dNTPs (Bioneer), 1 unit of Hot Start *Taq* DNA polymerase (Bioneer), 0.4 μM Atlantic cod primers, 1 μM Pacific cod primers, 0.6 μM blue whiting primers, 0.4 μM haddock primers, 0.28 μM Alaska pollock primers, and 10 ng template DNA. Multiplex PCR condition is identical with the simplex PCR procedure as mentioned above. All PCR amplicons obtained from the multiplex PCR were confirmed via capillary electrophoresis. 

## 3. Results and Discussion

### 3.1. Specificity and Sensitivity of Simplex PCR

The specificity of each simplex PCR was conducted using the DNA samples extracted from the 19 fish species, including Atlantic cod, Pacific cod, blue whiting, haddock, and Alaska pollock. As expected, species-specific primers amplified only the target species. PCR amplicon sizes were 237 bp, 204 bp, 164 bp, 138 bp, and 98 bp for Atlantic cod, Pacific cod, blue whiting, haddock, and Alaska pollock, respectively (Figure 1). The amplified DNA fragments were further confirmed by agarose gel, and the results were provided as supplementary data.

The sensitivity of each simplex PCR assay was evaluated using DNA diluted from 10 ng to 0.01 pg. The limit of detection (LOD) for the developed assay was 0.1 pg for Atlantic cod, Pacific cod, and Alaska pollock, whereas it was 1 pg for blue whiting and haddock (Figure 2). Compared with previous studies, our assay was more sensitive for detecting Atlantic cod, Pacific cod, and haddock [2,3]. Universally, real-time PCR and LAMP methods have been reported to be more specific and sensitive when compared with conventional PCR because of the fluorescent probes and several pairs of primers used during target detection [13,18,19,20]. However, the LOD of the assay designed in this study indicated values that ranged from 0.1 to 1 pg, which was highly sensitive compared with those obtained from other assays (20 pg for real-time PCR and 28.5–285 pg for LAMP), even though the assay is based on conventional PCR [2,3]. Therefore, the obtained results suggested that our designed primers can detect five *Gadidae* species even at a low concentration.

### 3.2. Specificity and Sensitivity of Multiplex PCR

To simultaneously detect Atlantic cod, Pacific cod, blue whiting, haddock, and Alaska pollock species, we developed a multiplex PCR assay coupled with capillary electrophoresis. We performed the preliminary experiments to determine the optimal concentration of each primer set (data not shown), then we found that the multiplex PCR condition (i.e., 0.4 μM/1.0 μM/0.6 μM/0.4 μM/0.28 μM for Atlantic cod, Pacific cod, blue whiting, haddock, and Alaska pollock, respectively) was most suitable without the cross-reactivity. With 19 fish species tested, the multiplex PCR assay showed no cross-reactivity (Figure 3). Furthermore, we verified the PCR results by analyzing agarose gel electrophoresis, and those data were provided as supplementary data.

The LOD of the multiplex PCR was defined when all five bands were detected. As shown in Figure 4, the LOD of multiplex PCR was 1 pg and it was repeated thrice on different days. When we looked at the individual targets, the sensitivity of each species was identical to that of the simplex PCR, even under multiplex PCR conditions. This result indicated that the designed primers did not inhibit each other in a single reaction and had high specificity, resulting in the sensitivity that corresponded to that of the simplex PCR. Compared to the LOD of similar studies of fish products (5 ng for tunas and billfishes, 1 ng for freshwater fish species, and 1 ng for puffer fish), the LOD of our multiplex PCR assay (1 pg) is very sensitive and can detect very small amounts of cod DNA [21,22,23]. These results suggest that this PCR assay can be applied to on-site detection with low DNA extraction efficiency or to fragmented DNA samples such as processed food.

The capillary electrophoresis technology provides better precision and resolution than agarose gel electrophoresis [24,25]. Capillary electrophoresis can also distinguish between similar sizes of PCR bands amplified with multiple primer sets more clearly, and even weak bands on the agarose gel can be confirmed through the peaks. Hence, we applied this technique to confirm each PCR amplicon and clearly distinguish between the different sizes of the bands through the peaks (Figure 4).

Although many reported detection methods exist for *Gadidae* family [2,3,26], most of them are simplex PCR techniques, which is more time-consuming and expensive to be used for identifying several cod species simultaneously. The DNA barcoding system is also among the most popular ways for the authentication of fish species; however, it requires another step after PCR, which is sequencing [27]. Alternatively, our multiplex PCR assay is an efficient detection method that saves time and running costs by identifying five *Gadidae* species in a single reaction. The multiplex PCR for Atlantic cod, Pacific cod, blue whiting, haddock, and Alaska pollock has not been reported yet. Thus, our assay can be useful as a rapid, accurate, and sensitive detection method of the five *Gadidae* species (i.e., Atlantic cod, Pacific cod, blue whiting, haddock, and Alaska pollock).

### 3.3. Monitoring

To evaluate the applicability of our multiplex PCR assay to processed food products, we tested 40 commercial products purchased from Korea, Russia, USA, Vietnam, China, Sweden, and Norway (Table 2). The application test was conducted using three different PCR machines in the laboratory to reproduce and validate our assay. As shown in Table 2, all commercial food samples showed positive results. These results indicated that our assay was applicable to various processed types of commercial food products containing degraded DNA. In particular, the sizes of all PCR amplicons in this study were less than 250 bp, suggesting that our assay is suitable even for processed products.

Codfish products usually have similar morphological and meristic characteristics after having completed the process of peeling, cutting, cooking, or canning [11]. Thus, visual classification is not noticeable, and fish products are susceptible to intentional or accidental substitution. Among the 40 tested food samples containing the target species, 39 samples were identical to the labeled information. However, one commercial sample was not matched with the labeled information. Thus, despite being labeled alone as Atlantic cod, it was adulterated with haddock (Table 2). To confirm PCR amplicons from the adulterated commercial sample, we further analyzed the sequences of PCR amplicons. Sequencing analysis was performed at GenoTech (Daejeon, Korea), and the sequences of PCR products were compared with Basic Local Alignment Search Tool (BLAST) of the NCBI database. The BLAST results indicated that the commercial sample was mixed with Atlantic cod and haddock (data not shown). Processed food products mixed with Atlantic cod and haddock have been reported in previous studies. For Atlantic cod, haddock was the most popular substitute species in France, Sweden, and Denmark [8]. Similar cases were found in Italy and Spain [7,28]. Mislabeled products associated with Atlantic cod and haddock were expected to be reported more because those two fishes were repeatedly caught in a mixed fishery [29]. Haddock was also listed as the frequently replaced cod species in the IUCN database [7,8]. Most cod products were labeled only “cod” without representing a specific species. The absence of standardized nomenclature for commercial seafood can lead to label fraud. Thus, it is necessary to mark it in detail, such as “Atlantic cod” or “Pacific cod” [2]. Additionally, although it was not confirmed in our study that Pacific cod and Alaska pollock was substituted with Atlantic cod, such seafood fraud was verified through previous studies [7,30]. These results showed that it was confirmed the seafood fraud cases of five cod species and the necessity for effective tools to manage labeling errors.

Molecular detection methods, especially PCR-based assays, are clearer and more accurate compared with other detection assays, as DNA remains after food processing. Our assay was therefore designed to be able to apply processed food products and was verified by different types of processed products, suggesting that the developed detection method can be applied to all processed food products tested. Thus, this highly sensitive and specific multiplex PCR method can detect five *Gadidae* species and can even detect further. Hence, it can be applied to various processed samples, and is useful for monitoring seafood adulteration.

The application test was conducted three times independently using different PCR instruments.

## 4. Conclusions

We designed each primer set to develop a multiplex PCR assay for identifying Atlantic cod, Pacific cod, blue whiting, haddock, and Alaska pollock, simultaneously. These primers were specific to those species and evaluated their sensitivities using capillary electrophoresis. The developed multiplex PCR method was unique for detecting the five cod species at once. Furthermore, fish authentication from these various types of processed food products was successful through this sensitive, efficient, and reliable method, suggesting that this assay can be used in the fish industry. Real-time PCR, as well as digital PCR methods can be applied to quantify the samples and to eliminate the electrophoresis step.

## Figures and Tables

**Figure 1 foods-10-02631-f001:**
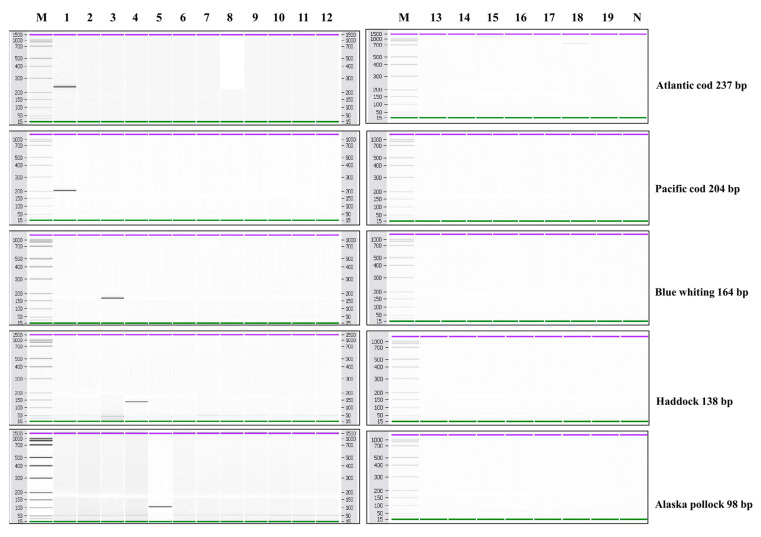
Specificity of the specific-species primers of Atlantic cod, Pacific cod, blue whiting, haddock, and Alaska pollock. Lane M: 100 bp DNA ladder, lane 1: Atlantic cod, lane 2: Pacific cod, lane 3: blue whiting, lane 4: haddock, lane 5: Alaska pollock, lane 6: common carp, lane 7: leather carp, lane 8: goldfish, lane 9: Chinese muddy loach, lane 10: snakehead, lane 11: Nile tilapia, lane 12: Pacific saury, lane 13: Pacific chub mackerel, lane 14: longtooth grouper, lane 15: convict grouper, lane 16: Atlantic salmon, lane 17: masu salmon, lane 18: swordfish, lane 19: Patagonian toothfish, lane N: non-template.

**Figure 2 foods-10-02631-f002:**
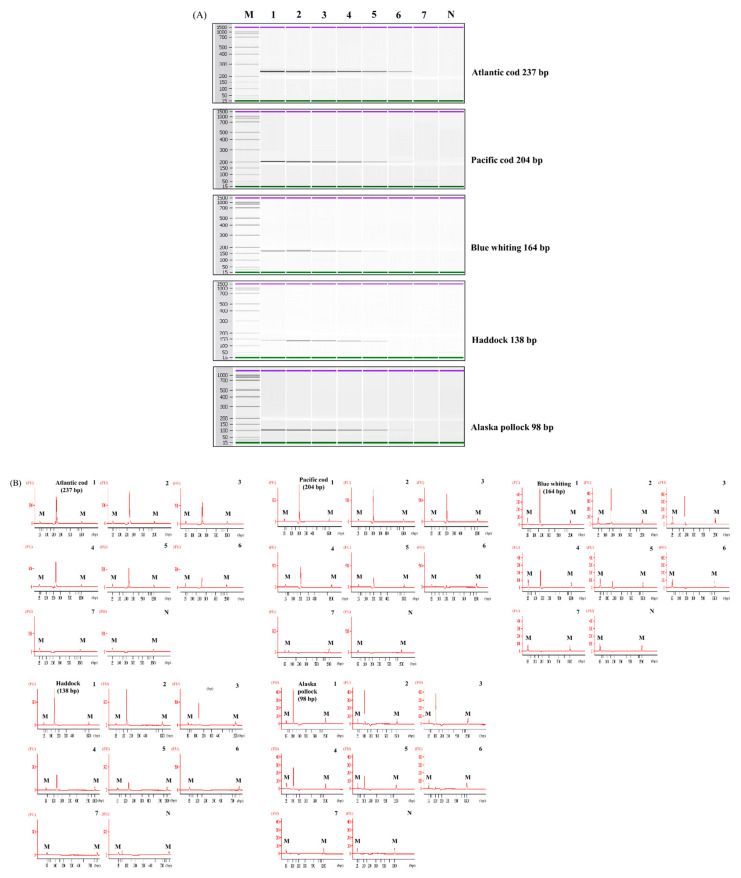
Sensitivity of the simplex PCR assay. (**A**) Gel, and Lane M: 100 bp DNA ladder, lanes 1–7: 10 ng to 0.01 pg of DNA from target species, and lane N: non-template. (**B**) Electropherogram, M: alignment marker, lanes 1–7: 10 ng to 0.01 pg of DNA from target species, and lane N: non-template.

**Figure 3 foods-10-02631-f003:**
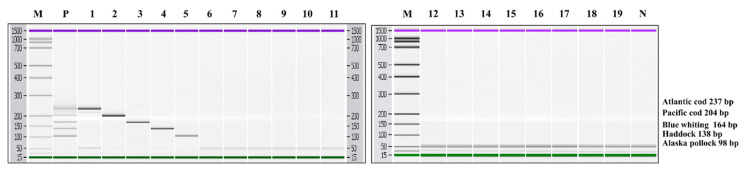
Specificity of the multiplex PCR assay. Lane M: 100 bp DNA ladder, lane P: positive control (10 ng of DNA from target species), lane 1: Atlantic cod, lane 2: Pacific cod, lane 3: blue whiting, lane 4: haddock, lane 5: Alaska pollock, lane 6: common carp, lane 7: leather carp, lane 8: goldfish, lane 9: Chinese muddy loach, lane 10: snakehead, lane 11: Nile tilapia, lane 12: Pacific saury, lane 13: Pacific chub mackerel, lane 14: longtooth grouper, lane 15: convict grouper, lane 16: Atlantic salmon, lane 17: masu salmon, lane 18: swordfish, lane 19: Patagonian toothfish, lane N: non-template.

**Figure 4 foods-10-02631-f004:**
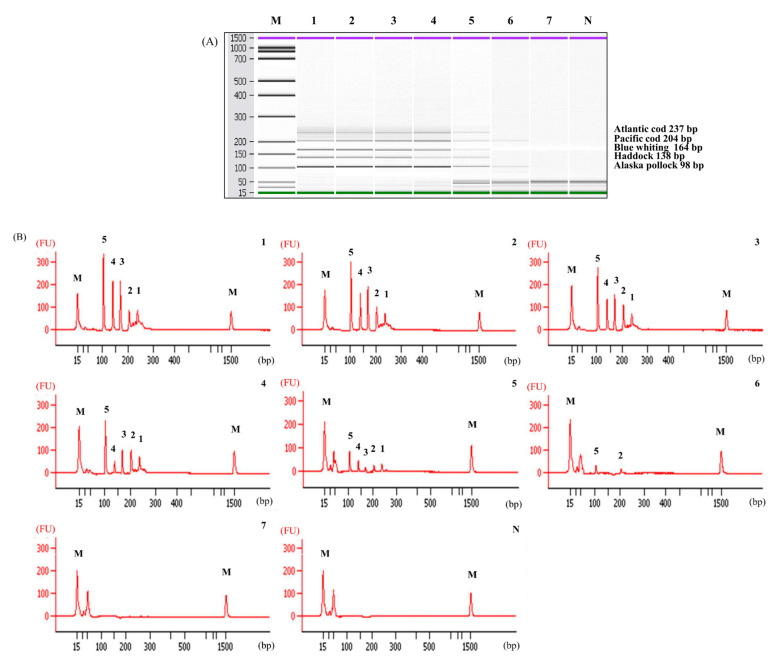
Limit of detection (LOD) of the multiplex PCR. (**A**) Gel, and Lane M: 100 bp DNA ladder, lanes 1–7: 10 ng to 0.01 pg of DNA from target species, and lane N: non-template. (**B**) Electropherogram, M: alignment marker, lanes 1–7: 10 ng to 0.01 pg of DNA from target species, and lane N: non-template. Numbers 1, 2, 3, 4, and 5 are Atlantic cod, Pacific cod, blue whiting, haddock, and Alaska pollock, respectively.

**Table 1 foods-10-02631-t001:** Primers used in this study.

Target Species	Primer Name	Sequence (5′→3′)	Target Gene	Amplicon Size (bp)	Reference
*Gadus morhua*	G.MOR_F	TCA ATG GAT CTG AGG AGG T	cyt b	237	This study
	G.MOR_R	GTT AAG CCC AGA AGC ATC			
*Gadus macrocephalus*	G.MAC_F	CAG TAG ATA ATG CCA CCT TG	cyt b	204	This study
	G.MAC_R	GAA GCA TTA CAG CAA AGC CG			
*Micromesistius poutassou*	M. POU_F	TCT CCT AGG CCT TTG CTT GG	cyt b	164	This study
	M. POU_R	AGA AAG AGG CAC CGT TAG CG			
*Melanogrammus aeglefinus*	M. AEG_F	GCA CTT GTT GAT CTT CCC AC	cyt b	138	This study
	M. AEG_R	GGC TGT TTC AAT GTC TGA AGT A			
*Gadus chalcogrammus*	G. CHA_F	CCC TTT CAC CCA TAT TTC ACG	cyt b	98	This study
	G. CHA_R	CCA AGC AAA TTA GGT GCG AAA			

**Table 2 foods-10-02631-t002:** Application and validation results of the multiplex PCR assay to commercial products.

No	Product Type	Labeled Species	Origin		Multiplex PCR Results			
				Atlantic Cod	Pacific Cod	Blue Whiting	Haddock	Alaska Pollock
1	Dried	Cod	Russia		+++			
2	Dried	Cod	Russia		+++			
3	Egg	Cod	USA		+++			
4	Egg	Cod	USA		+++			
5	Minced	Cod	Korea		+++			
6	Pancake	Cod	Russia		+++			
7	Pancake	Alaska pollock	Russia					+++
8	Dried	Alaska pollock	Russia					+++
9	Dried	Alaska pollock	Russia					+++
10	Roasted	Alaska pollock	Russia					+++
11	Dried	Alaska pollock	Russia					+++
12	Dried	Alaska pollock	Russia					+++
13	Salted	Blue whiting	Vietnam			+++		
14	Cutlet	Blue whiting	China			+++		
15	Cutlet	Blue whiting	China			+++		
16	Dried	Blue whiting	China			+++		
17	Fried	Alaska pollock	Russia					+++
18	Fried	Haddock	USA				+++	
19	Egg	Alaska pollock	USA					+++
20	Fried	Pacific cod	USA		+++			
21	Salted	Alaska pollock	Russia					+++
22	Salted	Alaska pollock	Russia					+++
23	Egg	Alaska pollock	Russia					+++
24	Egg	Atlantic cod	Sweden	+++			+++	
25	Fried	Alaska pollock	Russia					+++
26	Dried	Alaska pollock	Russia					+++
27	Dried	Alaska pollock	Russia					+++
28	Fried	Cod	Russia		+++			
29	Fillet	Cod	USA		+++			
30	Minced	Cod	Korea		+++			
31	Dried	Atlantic cod	Norway	+++				
32	Roasted	Atlantic cod	Norway	+++				
33	Boiled	Atlantic cod	Norway	+++				
34	Dried	Atlantic cod	Norway	+++				
35	Roasted	Atlantic cod	Norway	+++				
36	Boiled	Haddock	USA				+++	
37	Boiled	Haddock	USA				+++	
38	Roasted	Haddock	USA				+++	
39	Roasted	Haddock	USA				+++	
40	Dried	Haddock	USA				+++	

‘+’ means a positive result.

## Data Availability

The data presented in this study are available on request from the corresponding author.

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
