# Peer review of "A Multiplex PCR Assay Combined with Capillary Electrophoresis for the Simultaneous Identification of Atlantic Cod, Pacific Cod, Blue Whiting, Haddock, and Alaska Pollock"

_foods, 2021, doi:10.3390/foods10112631_

Round 1
Reviewer 1 Report
The method developed in the work is well designed and the results clearly show its effectiveness and reliability.
The work could be improved by DNA sequencing confirmation of the analyzed samples at the end of the work but overall the work is meritorious to be accepted after major revisions. The images of gel Electrophoresis are too much elaborated using an editing software. Using the original images at high resolution will improve the overall work scientific quality also if the images will be not so nice as the ones reported in the current version. Attached find the PDF file with 12 specific comments (activate comments).
Author Response
We are very thankful to the reviewers for their comprehensive and thorough review. Our detailed response to the review comments are given below (in blue). We have addressed all of the concerns by the reviewers and hope that the revised manuscript is now suitable for publication.
Reviewer #1
The method developed in the work is well designed and the results clearly show its effectiveness and reliability.
The work could be improved by DNA sequencing confirmation of the analyzed samples at the end of the work but overall the work is meritorious to be accepted after major revisions. The images of gel Electrophoresis are too much elaborated using an editing software. Using the original images at high resolution will improve the overall work scientific quality also if the images will be not so nice as the ones reported in the current version. Attached find the PDF file with 12 specific comments (activate comments).
L71, L72-77. Please check all the common names and use the ones accepted for FAO or fishbase.
Response: Thank you for your comments. We modified typing error and used the common names of all samples based on FAO and fishbase.
Line 71: Alaska pollock (Gadus chalcogrammus)
Lines 73-79: common carp (Cyprinus carpio), leather carp (Cyprinus carpio nudus), goldfish (Carassius auratus), Chinese muddy loach (Misgurnus mizolepis), snakehead (Channa argus), Nile tilapia (Oreochromis niloticus), Pacific saury (Cololabis saira), Pacific chub mackerel (Scomber japonicus), longtooth grouper (Epinephelus bruneus), convict grouper (Epinephelus septemfasciatus), Atlantic salmon (Salmo salar), masu salmon (Oncorhynchus masou), swordfish (Xiphias gladius), Patagonian toothfish (Dissostichus eleginoides)
L120-121. Why do you use different amount of primers in the multiplex recipe? It do not seems correlated with the primers size. Is there any preliminary experiment to determine the amount of each couple of primers?
Response: Multiple pairs of primers used in multiplex PCR can cause the competition between primer sets, thus, it may affect its specificity and sensitivity. It has been reported that the concentration of primers is necessary to be optimized to increase the sensitivity and minimize non-specific interactions (Cheng et al, 2016; Zha et al, 2010; Suh et al, 2020). Likewise, we adjusted the concentration of each primer set to optimize the multiplex PCR condition. Even though we did not mention the details of preliminary data in the main manuscript, we tested multiple conditions to determine the optimal concentration of each primer set. By testing different primer concentrations, we found the current condition (i.e., 0.4 μM/1.0 μM/0.6 μM/0.4 μM/0.28 μM for Atlantic cod, Pacific cod, blue whiting, haddock, and Alaska pollock, respectively) as the most suitable one.
As you suggested, we included the sentences in Line 163-167 to mention that the optimization of multiplex PCR was performed.
Lines 163-167: We performed the preliminary experiments to determine the optimal concentration of each primer set (data not shown), then we found that the multiplex PCRcondition (i.e., 0.4 μM/1.0 μM/0.6 μM/0.4 μM/0.28 μM for Atlantic cod, Pacific cod, blue whiting, haddock, and Alaska pollock, respectively) was the most suitable one without the cross-reactivity.
Figure 1, Figure 2(A). The images of gel Electrophoresis are too much elaborated using an editing software. It is not good to provide such images (it could be easily seen). Please provide the original image also if they are not so nice.
Response: Thank you for your comments. Figures 1 and 2(A) are the original images without editing software, and the images obtained by capillary electrophoresis are the default outputs from Bioanalyzer. We changed the figures with better resolution and additionally provided the supplementary data that analyzed by agarose gel to confirm our results. It was mentioned in lines 133-135.
Lines 133-135: The amplified DNA fragments were further confirmed by agarose gel, and the results were provided as supplementary data.
Figure 3. The images are elaborated and also coulored. Please to increase the strength of your work use high resolution images of your gels describing them into the legend and results test.
Response: Thank you for your suggestions. Figure 3 was edited with high resolution figures.
As mentioned earlier, we additionally confirmed the PCR amplicons by agarose gel electrophoresis, and the manuscript is revised (Lines 168-169).
Lines 168-169: Furthermore, we verified the PCR results by analyzing agarose gel electrophoresis, and those data were provided as supplementary data.
L205. It would be interesting to know if the DNA sequence of the resulting amplicon not matching with the label confirmed your method.
Response: As you recommended, we further confirmed the DNA sequences, and the results were added to the sentences in lines 220-225 as follows:
Lines 220-225: To confirm PCR amplicons from the adulterated commercial sample, we further analyzed the sequences of PCR amplicons. Sequencing analysis was performed at GenoTech (Daejeon, Korea), and the sequences of PCR products were compared with Basic Local Alignment Search Tool (BLAST) of the NCBI database. The BLAST results indicated that the commercial sample was mixed with Atlantic cod and haddock (data not shown).
L225. Maybe into the conclusions you could also give some future prospects of PCR method upgrade such as the use of a real-time PCR method to avoid the step of electrophoresis.
Response: As you recommended, we added the sentences in lines 250-252 as follows:
Lines 250-252: Real-time PCR as well as digital PCR methods can be applied to quantify the samples and to eliminate the electrophoresis step.
Reviewer 2 Report
Dear Madam
This manuscript is about a multiplex PCR method for the differentiation of five cod species. Food authenticity verification is a topic of interest for modern food production. This study is generally well planned, with interesting results concerning the method. The language should be proofread prior to acceptance by a native speaker since in several cases an edit is neeeded. Some points also need to be elucidated prior to final acceptance. The major drawback of this manuscript is that the discussion of the results is superficial, not discussing the results of this work with other researchers; therefore an extended revision is needed concerning the Results and Discussion part.
Specific points
P1 L21. Please rephrase.
P2 L74. “Chnna” is probably “Channa”.
P2 L77. According to fishbase, Oncorhynchus masou is Japanese salmon. Also, billfish is a broader term for Xiphias gladius. Swordfish is more precise.
P2 L84-85. How many different fishes from each species were sampled?
P5 Figure 2B. The image is of low resolution.
P6 L178-179. Please rephrase.
P7 Figure 3. The meaning of the grey areas should be mentioned.
P8 Table 2. No differentiation between one cross or three crosses. Since all groups produced the expected result, this table does not provide any information other than the product type and its origin.
Author Response
We are very thankful to the reviewers for their comprehensive and thorough review. Our detailed response to the review comments are given below (in blue). We have addressed all of the concerns by the reviewers and hope that the revised manuscript is now suitable for publication.
Reviewer #2
Dear Madam
This manuscript is about a multiplex PCR method for the differentiation of five cod species. Food authenticity verification is a topic of interest for modern food production. This study is generally well planned, with interesting results concerning the method. The language should be proofread prior to acceptance by a native speaker since in several cases an edit is neeeded. Some points also need to be elucidated prior to final acceptance. The major drawback of this manuscript is that the discussion of the results is superficial, not discussing the results of this work with other researchers; therefore an extended revision is needed concerning the Results and Discussion part.
Specific points
P1 L21. Please rephrase.
Response: As you recommended, we revised the sentence in lines 21-22 as follows:
Lines 21-22: The limit of detection of this multiplex PCR assay was 1 pg.
P2 L74. “Chnna” is probably “Channa”.
P2 L77. According to fishbase, Oncorhynchus masou is Japanese salmon. Also, billfish is a broader term for Xiphias gladius. Swordfish is more precise.
Response: As you recommended, we used the common names of all samples based on FAO and fishbase.
Lines 73-79: common carp (Cyprinus carpio), leather carp (Cyprinus carpio nudus), goldfish (Carassius auratus), Chinese muddy loach (Misgurnus mizolepis), snakehead (Channa argus), Nile tilapia (Oreochromis niloticus), Pacific saury (Cololabis saira), Pacific chub mackerel (Scomber japonicus), longtooth grouper (Epinephelus bruneus), convict grouper (Epinephelus septemfasciatus), Atlantic salmon (Salmo salar), masu salmon (Oncorhynchus masou), swordfish (Xiphias gladius), Patagonian toothfish (Dissostichus eleginoides)
P2 L84-85. How many different fishes from each species were sampled?
Response: Five cod species were obtained from the National Institute of Food and Drug Safety Evaluation as a single reference. And, 14 non-target species were purchased from three domestic markets and mixed for use. Forty commercial products were purchased as a single item. This information was updated in 2.1. Samples section under Materials and Methods.
P5 Figure 2B. The image is of low resolution.
Response: As you recommended, Figure 2B was edited with high resolution.
P6 L178-179. Please rephrase.
Response: As you recommended, we revised the sentence in lines 186-188 as follows:
Lines 186-188: On the other hands, our multiplex PCR assay is an efficient detection method that saves time and running costs by identifying 5 Gadidae species in a single reaction.
P7 Figure 3. The meaning of the grey areas should be mentioned.
Response: The grey areas of previous Figure 3 was the background of each lane. We used the default setting in Bioanalyzer to make the brightness of background images were identical, and the figure was updated (Figure 3). Bioanalyzer results were further confirmed through agarose gel electrophoresis as shown in the supplementary data.
P8 Table 2. No differentiation between one cross or three crosses. Since all groups produced the expected result, this table does not provide any information other than the product type and its origin.
Response: Thank you for your comments. Table 2 indicates the application test to commercial food products as well as the validation data to confirm the reliability of this assay. We conducted these experiments with three different PCR machines in the laboratory to reproduce and validate our assay. To better explain the meaning of Table 2, we revised the title as ‘Application and validation results of the multiplex PCR assay to commercial products’. Also, we updated the legend in line 242 as follows:
Line 242: The application test was conducted three times independently using different PCR instruments.
Round 2
Reviewer 1 Report
The authors improved the work after the first review.
Only the advice to remove the electrophoresis images that appeared too edited was not accepted, however they correctly inserted them in their original form in the supplementary materials. The work is worthy of publication in its current form. A final revision of the English, maybe during proofreading, would be desirable to increase readability.
Author Response
Thank you for your review. We added the result of agarose gel electrophoresis in the supplementary materials.